# Regenerative and Transplantation Medicine: Cellular Therapy Using Adipose Tissue-Derived Mesenchymal Stromal Cells for Type 1 Diabetes Mellitus

**DOI:** 10.3390/jcm8020249

**Published:** 2019-02-15

**Authors:** Hiroyuki Takahashi, Naoaki Sakata, Gumpei Yoshimatsu, Suguru Hasegawa, Shohta Kodama

**Affiliations:** 1Department of Regenerative Medicine & Transplantation, Faculty of Medicine, Fukuoka University, 7-45-1 Nanakuma, Jonan-ku, Fukuoka 814-0180, Japan; md150011@cis.fukuoka-u.ac.jp (H.T.); gyoshimatsu@fukuoka-u.ac.jp (G.Y.); skodama@fukuoka-u.ac.jp (S.K.); 2Center for Regenerative Medicine, Fukuoka University Hospital, 7-45-1 Nanakuma, Jonan-ku, Fukuoka 814-0180, Japan; 3Department of Gastroenterological Surgery, Faculty of Medicine, Fukuoka University, 7-45-1 Nanakuma, Jonan-ku, Fukuoka 814-0180, Japan; shase@fukuoka-u.ac.jp

**Keywords:** type 1 diabetes mellitus, adipose tissue-derived mesenchymal stromal cell, insulin-producing cell, islet transplantation, differentiation, immunomodulation, revascularization

## Abstract

Type 1 diabetes mellitus (T1DM) is caused by the autoimmune targeting of pancreatic β-cells, and, in the advanced stage, severe hypoinsulinemia due to islet destruction. In patients with T1DM, continuous exogenous insulin therapy cannot be avoided. However, an insufficient dose of insulin easily induces extreme hyperglycemia or diabetic ketoacidosis, and intensive insulin therapy may cause hypoglycemic symptoms including hypoglycemic shock. While these insulin therapies are efficacious in most patients, some additional therapies are warranted to support the control of blood glucose levels and reduce the risk of hypoglycemia in patients who respond poorly despite receiving appropriate treatment. There has been a recent gain in the popularity of cellular therapies using mesenchymal stromal cells (MSCs) in various clinical fields, owing to their multipotentiality, capacity for self-renewal, and regenerative and immunomodulatory potential. In particular, adipose tissue-derived MSCs (ADMSCs) have become a focus in the clinical setting due to the abundance and easy isolation of these cells. In this review, we outline the possible therapeutic benefits of ADMSC for the treatment of T1DM.

## 1. Introduction: The Current Status of Type 1 Diabetes Mellitus

Diabetes mellitus (DM) is a common disease that has rapidly increased worldwide [1]. In 2015, it was estimated that 415 million people were living with DM, with this prevalence predicted to rise to 642 million by 2040 [1]. Long-term DM is associated with various comorbid conditions including neuropathy, retinopathy, nephropathy, and ischemic heart disease, which often impair patient quality of life (QOL) [2,3]. Type 1 diabetes mellitus (T1DM), which counts for approximately 5% to 10% of DM cases, is caused by an autoimmune response against pancreatic β-cells [4,5]. Most of the pancreatic islets are destroyed and, in advanced stages, patients present with severe hypoinsulinemia [4,6]. These patients require continuous exogenous insulin therapy, with insufficient doses leading to extreme hyperglycemia or diabetic ketoacidosis. In some cases, intensive insulin therapy may further result in hypoglycemic symptoms including hypoglycemic shock [7]. The recent remarkable development of insulin therapies including various types of insulin agents and insulin pumps, have allowed patients to properly and easily control their blood glucose levels [8,9]. In particular, recent randomized trials showed that patients who used insulin pump therapy had better improvements in hemoglobin A1c (HbA1c) levels and QOL as compared with patients treated with multiple daily insulin injections. Despite this, the risk of severe hypoglycemia is similar between both groups [10]. Yet, although most patients with T1DM can control their blood glucose levels through insulin therapies, there is a need for some additional therapies to support the control of blood glucose levels, minimize the risk of hypoglycemia, and help patients recover their blood glucose control in patients who struggle to maintain healthy levels despite receiving a consistent insulin therapy.

One of the effective surgical therapy for T1DM is pancreas transplantation. Pancreas transplantation can rescue T1DM patients with renal disease in end stage and unaware hypoglycemia. However, this therapy presents post-operational risk for complications such as thrombosis, pancreatitis, infection bleeding and rejection [11]. Another possible therapy is islet transplantation, a type of cellular replacement therapy that can rescue the production of an appropriate amount of insulin in a glucose-dependent manner and thereby diminish the risk of hypoglycemia [12,13]. This therapy would also enable patients to be free from the need to administer insulin [14,15]. However, compared with pancreas whole organ transplantation, islet transplantation has poor efficacy. Furthermore, despite the development of a method for systematic islet isolation [16] and the recent use of an immunosuppression protocol [12], it remains difficult to engraft transplanted islets with good long-term survival [17,18]. Thus, alternative therapies to protect resident cells that also possess physiological, insulin-releasing potential over the long term are desired. We review herein the possibility of an alternative therapy that is stem cell therapy.

Cellular therapy using mesenchymal stromal cells (MSCs) has gained popularity in recent years across various clinical fields [19,20]. Owing to its multipotentiality, its capacity for self-renewal, and its regenerative and immunomodulatory potential, MSCs may prove a useful cellular resource for treating T1DM. In particular, adipose tissue-derived MSCs (ADMSCs) have been targeted as potential therapy cells due to their abundance and easy isolation. In this review, we show the possibilities of ADMSC therapies for T1DM.

## 2. What are Adipose Tissue-Derived Mesenchymal Stromal Cells?

### 2.1. Properties of Mesenchymal Stromal Cells

The stem cell is the root of all mature cellular lineages and possesses self-renewal potential as well as the capacity to differentiate into multiple functional cells [21]. MSCs are multipotent stromal cells that can differentiate into endothelial cells [22], cardiomyocytes [23], hepatocytes [24] or neural cells [25]. MSCs also release various growth and inflammatory factors, such as vascular endothelial growth factor (VEGF), hepatocyte growth factor (HGF), fibroblast growth factor (FGF) and prostaglandin E2 (PGE2) [26] (Figure 1) to contribute to the repair of injured tissues [26,27]. Moreover, MSCs are excellent immunomodulators, able to manage severe inflammation and the immune system by suppressing T cell activation, proliferation and maturation [28,29]; inhibiting naïve and memory T cell response [29]; and upregulating the production of regulatory T cells (Tregs) [30]. In the early 1970s, Friedenstein and his colleagues first identified fibroblastic precursor cells in the murine bone marrow [31]. These cells were named as MSCs in the 1980s [32]. MSCs are adult stromal cells derived from the mesoderm and part of the ectodermal neural crest [33,34]. In the clinical setting, MSCs are well tolerated, and their use is less associated with ethical concerns as compared with embryonic stem cells (ESCs) [35].

According to the Mesenchymal and Tissue Stem Cell Committee of the International Society for Cellular Therapy [36], MSCs are defined by three parameters: the cells (1) are plastic-adherent and can be maintained in standard culture conditions; (2) express cluster of differentiation (CD)-105, CD73 and CD90 without hematopoietic markers (e.g., CD45, CD34, CD14, CD11b, CD79α, CD19 or HLA-DR); and (3) differentiate into osteoblasts, adipocytes, and chondroblasts in vitro. Previously, it was considered that MSCs could be isolated from mesodermal tissues only. However, more recent studies have identified MSCs in almost all tissues or postnatal organs including skeletal muscle, skin, placenta, umbilical cord, dental pulp, thymus, liver, adrenal grand, pancreas, spleen, and adipose tissue [20,37,38,39,40,41,42,43,44,45,46].

Adipose tissue plays important roles in energy storage, insulation, and as a producer of numerous endocrine mediators, such as adipokines or lipokines [47]. These roles are carried out by adipocytes, the major cellular component of adipose tissue. Although few in number, stromal cells within adipose tissue are responsible for replenishing mature adipocytes. These ADMSCs retain the capacity for self-renewal and differentiation throughout the organism’s lifetime [48], and possess the capacity for homing, immunomodulation, promotion of repair, and direct regeneration of damaged tissues [49,50,51]. Like stromal cells in other organs, ADMSCs can differentiate into various cells like other MSCs including osteogenic, adipogenic, myogenic, and chondrogenic lineages.

### 2.2. Characteristics of Adipose Tissue-Derived Mesenchymal Stromal Cells

Despite their similarity to bone marrow-derived (BM) MSCs, ADMSCs have some unique characteristics. First, is their unique phenotypic pattern of expression. Whereas both MSCs and ADMSCs are positive for CD29, CD44, CD71, CD90, CD105/SH2 and SH3, only ADMSCs express CD36 and CD49d, and CD106 is common only to BM-MSCs [46,52,53]. These phenotypic expression patterns might be associated with differences in the derivation of ADMSCs and BM-MSCs [46,54].

The second unique feature is in its multipotency, with differences in the polarity, capacity, and kinetics between the two types of MSCs [46]. ADMSCs have a significantly higher proliferative and differentiative potential than BM-MSCs, whereas BM-MSCs show higher capacities for osteogenesis and chondrogenesis [53,55].

The third unique feature is in its immunomodulatory abilities. Although more than 90% of the immune-phenotypes are similar between ADMSCs and BM-MSCs [46,56,57], recent evidence indicates that ADMSCs are stronger immunomodulators than other types of MSCs. For example, Blancher and colleagues showed that ADMSCs had no capacity to strengthen the alloreactivity of lymphocytes, but suppressed mixed lymphocyte reactions and lymphocyte proliferative responses [56]. Moreover, Camara’s group found that ADMSCs promote the expression of programming death ligand-1 (PD-L1), which induces the expansion of CD4+Foxp3+ cells (Tregs) and inhibits the proliferation of lymphocytes [58,59]. In addition, ADMSCs can functionally inhibit the expression of CD80, CD83 and CD86 that are important molecules on dendritic cells (DCs) more effectively than BM-MSCs [60]. Collectively, these functions imply the stronger immunomodulatory functions of ADMSCs and their usefulness for clinical transplantation.

The fourth attribute of ADMSCs is their anti-inflammatory function and trophic effects. ADMSCs produce various pro-inflammatory and anti-inflammatory cytokines [61,62,63]. Among these cytokines, ADMSCs produce significantly higher amounts of interleukin (IL)-1Ra [61], IL-6 [61], interferon (IFN)-γ [64] and transforming growth factor (TGF)-β [63,65]; and lower volumes of IL-12 [66] as compared with BM-MSCs. ADMSCs also release higher amounts of growth factors including granulocyte colony stimulating factor (G-CSF) [61], granulocyte macrophage colony stimulating factor (GM-CSF) [61], VEGF [65,67,68], HGF [61,67,68], keratinocyte growth factor (KGF) [65], insulin-like growth factor 1 (IGF-1) [64], and FGF [62,64,65].

In summary of this chapter, ADMSCs have some superior functions to other MSCs in terms of their immunomodulatory, anti-inflammatory, and trophic effects. These advantages might contribute to the beneficial effects of ADMSC transplantation in patients with T1DM. Their superior immunomodulatory and anti-inflammatory functions may contribute to prevention of graft rejection, and the better trophic effect support engraftment of transplanted islets.

## 3. Cellular Therapy Using Adipose Tissue-Derived Mesenchymal Stromal Cells for Insulin-Dependent Type 1 Diabetes Mellitus

### 3.1. Outline of Cellular Therapy Using Adipose Tissue-Derived Mesenchymal Stromal Cells

The initial research on the use of MSCs for T1DM commenced in 2003, anticipating tolerance and neogenesis of the resident pancreatic islets [69,70]. Numerous studies have also been promoted since the discovery of insulin-producing cells (IPCs) derived from adipose tissue, also first published in 2003 [71]. The expected therapeutic effects of cellular therapy using ADMSCs can be roughly classified into the following four categories: (1) the use of differentiated-IPCs for transplantation; (2) methods to support and improve the function and proliferation of resident pancreatic islets; (3) methods that support the engraftment of co-transplanted islet grafts; and (4) methods that support the function of cultured islet grafts for transplantation.

### 3.2. Differentiated, Insulin-Producing Cells for Transplantation

Over the years, several types of stromal cells, including ADMSCs, have been exploited for the potential to derive IPCs [72,73]. Table 1 outlines the efficacies of differentiated-IPCs derived from ADMSCs [74,75,76,77,78,79,80,81,82,83]. The first trial for the differentiation of IPCs from human ADMSCs was performed by Timper and colleagues [74]. ADMSCs were isolated from human adipose tissues harvested during plastic surgery. The cells were then cultured in serum-free medium supplemented with exendin-4, pentagastrin, activin-A, betacellulin, nicotinamide, and HGF (Figure 2). The cells expressed pancreatic endocrine cellular phenotypes and islet genes, such as insulin, glucagon and somatostatin, which were evidence of their differentiation into IPCs. However, the potential for insulin secretion was not shown. Several years later, Kang et al. succeeded in differentiating human ADMSCs into IPCs using glucagon-like peptide-1 (GLP-1). They detected that the IPCs released insulin and C-peptide in a glucose-dependent manner. Moreover, they transplanted 1.5 × 10^6^ differentiated IPCs into the renal subcapsular space of diabetic mice and found an increase in serum insulin levels and achieved normoglycemia. This therapeutic effect was brought about by the transplanted IPCs rather than the regeneration or the repopulation of endogenous β-cells [76]. This is the first study to develop IPCs that can work as endocrine hormone-producing cells. Amer’s group also showed similar findings in a rat model [83].

Mature, differentiated IPCs from ADMSCs phenotypically express Pdx1 [77,78,84], MafA [85], Nkx2.2 [85], Nkx6.1 [85], Ngn3 [74,78,84,85], NeuroD [78], Pax-4 [78], Isl1 [74,85], Ipf-1 [74] and insulin [85]. Various factors contribute to IPC differentiation. The Wnt signaling pathway is one of the best characterized pathways, strongly correlated with many biological processes, including proliferation, apoptosis, and differentiation [86]. It also plays an important role in pancreas development, islet function, and insulin production and secretion [87,88]. Wang and colleagues showed that activation of Wnt signaling induced IPC differentiation from rat ADMSCs, identified through the detection of specific markers for IPCs, such as insulin, PDX1, and glucagon genes, and the protein expression of PDX1, CK19, nestin, insulin, and C-peptide [89]. The phosphoinositide-3 kinase (PI3K)/Akt signaling pathway is another important pathway involved in IPC differentiation. Tarique’s and Anjum’s groups have revealed that the PI3K/Akt signaling pathway is active during the development of IPCs from ADMSCs mediated by stromal cell-derived factor 1α (SDF-1α; also referred to as the CXCL12 chemokine) and basic fibroblast growth factor (bFGF) [90]. A recent study showed that overexpression of microRNA-375 is also important in the development of IPCs from ADMSCs [91]. mRNA-375 is correlated with insulin secretion [92] and β-cell proliferation [93]. Finally, the sonic hedgehog (Shh) signaling pathway is also necessary for the development of IPCs. Dayer et al. revealed that inhibition of the Shh pathway must be removed for IPC development [85].

As a donor source of IPCs, ADMSCs are not inferior to BM-MSCs. At least, there is no prominent difference between IPCs derived from BM-MSCs and ADMSCs in terms of the potential for insulin release or C-peptide production in response to glucose administration [94,95]. Furthermore, the insulin-releasing capacity of both derivatives of MSCs are reinforced when co-cultured with islet grafts [95].

Most of the studies involving ADMSC transplantation have used IPCs differentiated from ADMSCs. Some groups have attempted to clarify the therapeutic effects of undifferentiated ADMSC transplantation, but the benefits appear to be limited. Although Chandra et al. showed similar transplant outcomes between undifferentiated-ADMSCs and differentiated-IPCs in streptozotocin (STZ) treated-mice [78], many other studies have failed to achieve normoglycemia in their transplantations with undifferentiated-ADMSCs alone [76,77,80,82]. In addition, the immunomodulatory properties of ADMSCs can be maintained during the differentiation process [96]. This means that differentiated-IPCs may be tolerant of severe graft rejection. Taken together, differentiated-IPCs offer a reasonable cellular resource for transplantation as compared with undifferentiated-ADMSCs.

### 3.3. Functional Role of Adipose Tissue-Derived Mesenchymal Stromal Cells in the Resident Pancreatic Islets

ADMSCs are not only the donor source of IPCs, but also likely support resident pancreatic islets as well as local BM-MSCs [69]. Kono et al. infused STZ-treated, diabetic, immunodeficient mice with human ADMSCs and found an increase in serum insulin levels and improved glucose tolerance. The authors also found that these transplanted ADMSCs released cytokines, including interferon-inducible protein 10 (IP-10), eotaxin, VEGF, and tissue inhibitor of metalloproteinase 1 (TIMP-1), which all contribute to β-cell proliferation and the prevention of β-cell death [97] (Figure 3). Bassi’s group also showed that allogeneic ADMSC transplantation improved hyperglycemia in early-onset autoimmune diabetes by attenuating the Th1-related immune response and inducing the expansion of Tregs in nonobese diabetic (NOD) mice [59].

While the usefulness of ADMSC transplantation in diabetic animal models is recognized, there are some studies that have failed to ameliorate hyperglycemia by this treatment [76,77,80,82,98]. One possibility for this unexpected outcome is the difference in the numbers of transplanted ADMSCs. Dang and colleagues indicated that a measurable number of ADMSCs were required to support resident islets and enable the proper control of blood glucose levels [99]. Another possibility is the time of transplantation with respect to disease progress. Indeed, approximately 70% to 90% of pancreatic β-cells are destroyed in patients with T1DM by the time these patients receive their initial diagnosis [4,6]. Thus, it may be difficult to recover a sufficient number of resident islets after a certain disease threshold in T1DM. We consider that the application of ADMSC therapy for preserving resident islets should be limited to children with an early stage of onset and who show a measurable number of ADMSCs for therapy. However, an adequate supplementation of differentiated-IPCs may be required to treat adult patients with T1DM.

### 3.4. Supporting the Function and Engraftment of Co-Transplanted Islet Grafts

As mentioned previously, islet transplantation is available as a cellular replacement therapy for the recovery of endocrine function. However, in patients with T1DM, the number of vascular endothelial cells are reduced, which affects the potential for neovascularization [100]. This is one of the major factors contributing to the limited success of islet engraftment. As such, combinatorial approaches have been examined to boost the regenerative potential of MSCs, such as combining islet transplantation with MSCs transplantation (i.e., hybrid islet transplantation) [20]. Though positive outcomes have been shown for several studies involving hybrid islet transplantation using BM-MSCs, the effectiveness of such an approach is limited [101,102,103,104].

Over the past 10 years, many groups have attempted to clarify the utility of hybrid islet transplantation using ADMSCs in terms of its efficacy and versatility [95,98,105,106,107]. The first study was published in 2010. Ohmura et al. transplanted murine islet grafts with syngeneic ADMSCs into STZ-induced diabetic mice, and found that the treatment could reverse diabetes, with prolonged graft survival. Significant angiogenesis and a marked inhibition of inflammatory cell infiltration were also noted [105]. Several other studies also showed the usefulness of ADMSC co-transplantation, as outlined in Table 2.

ADMSCs presumably contribute to the available outcomes of hybrid islet transplantation in three ways. The first is through neovascularization. ADMSCs promote the establishment of a neovascular network by secreting various pro-angiogenic factors, including VEGF [108,109], HGF [108], kinase insert domain receptor (KDR) [108], TGF-β [110], and IL-8 [111] (Figure 4). The second way is through the prevention of inflammation. In animal models [98,112,113], ADMSC have been shown to significantly reduce the expression levels of proinflammatory cytokines such as TNF-α [98,113], IFN-γ [110], IL-6β [98], and IL-17 [110]. In vitro, ADMSCs suppressed the production of IFN-γ, IL-2, and IL-17 from diabetes mice-derived lymphocytes [112]. The third way is through controlling immunity. Whereas ADMSCs inhibit the infiltration of CD4+ and CD8+ T cells [105] and macrophages [113], they promote the production and infiltration of Tregs into the transplant site [107].

These transplant efficacies are reinforced through the addition of FGF-2 [109]. Likewise, factors that help to promote the expansion of the transplanted islets and enhance the expression of endocrine function may also sustain the efficacy of hybrid islet transplantation. For example, Tanaka and colleagues revealed that islet grafts were expanded in hybrid islet transplantation [106]. Karaoz’s group showed that co-transplanted ADMSCs also differentiate into IPCs [95], whereas Song and colleagues suggested that IGF-1 released by ADMSCs enhanced the survival of co-transplanted islet grafts [113].

Although there are only a few reports about the efficacy of hybrid islet transplantation directly using ADMSCs, the approaches using ADMSCs are likely to be superior to using BM-MSCs in terms of co-transplanted islet engraftment and function [95]. Given the numerous advantages associated with the use of ADMSCs, it may be reasonable to shift away from using BM-MSCs for hybrid islet transplantation and focus on optimizing the use of ADMSCs.

### 3.5. In Vitro Co-Culture of Syngeneic Islets Graft with Adipose Tissue-Derived Mesenchymal Stromal Cells Ameliorates Transplantation Efficiency

During the process of isolation and culture for islet transplantation, some of the pancreatic islet grafts are lost due to the hypoxic stress, poor blood supply, and the expression of inflammatory cytokines [110,114]. Thus, the loss of islet grafts during the harvesting process remains a major challenge.

MSCs have the potential to repair damaged tissue at the cellular level [14,115], and therefore, the cultivation of islet grafts with MSCs before transplantation could help to minimize graft loss. To this end, various co-culture experiments combining mouse or human islets with ADMSCs have been conducted in vitro [95,97,98,109,110,116]. Rackham’s group examined the therapeutic effect of co-culturing ADMSCs with syngeneic isolated islets in vitro, and verified that the insulin-releasing function of these islets was significantly better than that measured for islets cultured alone. They further showed that transplantation of these co-cultured islet grafts resulted in successful engraftment and a significant improvement in hyperglycemia in diabetic mice as compared with islets grafts not precultured with ADMSCs [116]. These improved therapeutic effects appear to be associated with paracrine crosstalk between the co-cultured islets and ADMSCs, leading to an upregulation of eotaxin [97], VEGF [97,109], TIMP-1 [97], extracellular matrix (ECM) components, annexin A1 (ANXA1) [110], and FGF-2 [109], all of which are enhanced when ADMSCs are cultured under hypoxic conditions [109,117,118]. Overexpression of betatrophin gene similarly improves islet viability when co-cultured with ADMSCs (Figure 5) [82].

Compared with BM-MSCs, ADMSCs are more likely to better support precultured islet grafts in terms of viability, recovery rate [119], and recovery of insulin-producing function [95]. Thus, ADMSCs serve as a reasonable support for the reinforcement of impaired islet grafts even during a hypoxic procedure like islet graft isolation. In summary, preculturing islet grafts with ADMSCs can contribute to the success of hybrid islet transplantation [95,98,110].

## 4. Towards a Suitable Adipose Tissue-Derived Mesenchymal Stromal Cells

### 4.1. Sources of Adipose Tissue-Derived Mesenchymal Stromal Cells

Adipose tissue is classified into main two categories: brown and white adipose tissues [120]. Brown adipose tissue is localized in cervical, paravertebral, supraclavicular, axially and suprarenal tissues, whereas white adipose tissue is found in craniofacial, pericardial, perirenal, omental, abdominal, and intestinal regions, as well as in the buttocks, thighs, and bone marrow [62]. White adipose tissue is thought to have the highest proportion of MSCs, with both of the number and differentiation potential of MSCs lower in brown adipose tissue sources [120].

ADMSCs are stored in various repositories. Subcutaneous depots have gained particular interest in terms of availability, abundance, and renewability [121,122,123]. The lower abdomen and the inner thigh are likely to contain high processed lipoaspirate cell concentrations [121,122,123,124]. In addition, ADMSCs in the subcutaneous abdominal wall tend to show higher differentiative potential in adipogenic and osteogenic cultures, as compared with cells from intra-abdominal adipose tissue sources (including the omentum and intestines) [123,125]. In terms of stem cell recovery, a higher number of viable cells are recovered from the subcutaneous adipose tissue from the arm as compared with tissues of the thigh, abdomen, and breast [48].

### 4.2. Method for the Preparation of Adipose Tissue-Derived Mesenchymal Stromal Cell

Compared with other stromal cells, ADMSCs tend to be in abundant supply and are easily harvested. In addition, large numbers of ADMSCs can be acquired from adipose tissue without any severe complications [126,127].

In the clinical setting, ADMSCs are usually isolated according to the following procedure: (1) The subcutaneous adipose tissues are collected by lipoaspiration; (2) The tissues are washed, digested using collagenase, and collected as a pellet through centrifugation; (3) The pellet—referred to as the stromal vascular fraction, and including blood cells, fibroblasts, pericytes and endothelial cells—is cultured. ADMSCs are collected through selection [48,128]. Approximately 5000 ADMSCs are collected per 1 gram of adipose tissue without risks of severe complication. This is approximately 500-times the yield of stromal cells attained from the bone marrow [129].

### 4.3. Various Factors Influencing the Quality of Adipose Tissue-Derived Mesenchymal Stromal Cells

An autologous cell source is an ideal choice for cellular therapy. However, it is difficult to use autologous ADMSCs in patients with severe illness who may have impaired ADMSC function (so-called pathologic ADMSCs) or an urgent necessity for ADMSCs. Thus, the establishment of an ADMSC bank system as a source of quality-controlled ADMSCs is desirable. For the quality control of ADMSCs stock, it is important to disclose donor characteristics, such as information about age, gender, and past and current illnesses (including DM and obesity), as well as the donor source location (subcutaneous tissue, omentum, for example) and type of adipose tissue (white or brown). These factors may affect cellular differentiation, multipotency, immunomodulation, and in some cases, malignant potential.

The influence of donor age on the quality of ADMSCs remains unclear. Some groups have asserted that there is no correlation between donor age and the quality of ADMSCs [121,124,130,131,132], while others suggest that ADMSCs harvested from older persons have lower proliferative capacity and multipotentiality [133,134,135,136]. Regarding gender, there is no evidence to support a functional difference between male and female donors. Lipoaspiration therapies tend to be performed in female patients undergoing plastic surgery [137].

Concerning body mass index (BMI), some groups report no or a negative correlation between ADMSC yield (per a milliliter of liposuction) and patient BMI [124,130,131]. Thus, there is no positive correlation between the total numbers of acquired ADMSCs and BMI. However, van Harmelen and colleagues revealed that the total number of adipocytes and stromal cells increased with higher BMI, and that cellular size was also enlarged with a higher BMI [131]. However, other studies show impaired self-renewal and differentiation abilities of ADMSCs from donors with higher BMI values, surmised to be due to the changes in telomerase activity [138] or mitogen-activated protein 4 kinase (MAP4K) expression [139].

Regarding metabolic syndrome, one study revealed that ADMSCs from donors without metabolic syndrome were more effective promoters of revascularization than those from donors with metabolic syndrome in terms of the normal angiogenetic and anti-inflammatory potential of the cells [140]. Hyperglycemia in DM impairs the angiogenetic potential of ADMSCs [135,141]. Koci and others claimed that adipose tissues from donors with DM were not satisfactory as an autologous source of ADMSCs as compared with tissues from donors without DM, with the cells performing inferiorly, both phenotypically and functionally [142]. However, Yaochite’s group suggested that BM-MSCs isolated from early-diagnosed patients with T1DM were not phenotypically or functionally impaired, indicating that the ADMSCs derived from newly diagnosed patients might have preserved multi-potency [143].

Taken together the above-described observations indicate that subcutaneous white adipose tissue from the abdominal wall of healthy, non-obese, young donors are an ideal source of ADMSCs. However, given that ADMSCs and BM-MSCs can significantly vary between donors, the similarities and differences in the properties of MSCs should be taken into consideration in these stem cell-based therapies [53].

### 4.4. The Origin and Malignant Potential of Adipose Tissue-Derived Mesenchymal Stromal Cells

MSCs exist in various tissues, including adipose tissue, and the origin of ADMSCs remains unclear. Some studies have proposed that MSCs originated from perivascular cells, such as pericytes [144,145,146,147], whereas others have suggested that they are scattered in fat stroma [148]. It is important to identify the origin of ADMSCs, because stem cells possess the potential for malignant transformation; albeit, at a very low frequency in humans [149]. At present, cancer therapeutic strategies depend on the genesis and stage [150]. Therefore, we should identify the origin, and prepare for the indeterminate occurrence of stromal cell-derived neoplasms.

Yet, to the best of our knowledge, there have been no reports of malignancy after MSC transplantation in human clinical situations [20]. However, in experimental cultures, malignant mutations arise from an accumulation of genomic alterations that occur following long-term self-renewal or repeated passaging [151,152]. Therefore, the optimal numbers of cellular passages for ADMSC induction should be investigated to maintain the multipotentiality of the cells, and prevent malignant activation. Among human ADMSCs, cells up to passage 10 appear to retain almost all the characteristics and properties of cells of earlier passage numbers [153,154]. And, although no DNA fragmentation has been found in vitro after expansion past 10 passages, the safety (i.e., chromosome stability) of ADMSCs cannot be ensured [155]. Prolonged ADMSCs passaging, such as repeating passaging for more than 4 months, has been linked with malignant transformation [156]. Physiological stress and in vitro culture conditions may also cause cellular or chromosomal abnormalities [157].

One study suggested that ADMSCs lacked telomeric activation, which might suppress the malignant potential of transplanted MSCs [158]. However, several studies have reported the direct contribution of ADMSCs to malignant transformation and tumor growth in some types of neoplasms [159,160,161,162]. Further studies are required to determine the potential impact of stem cell therapies on malignancy.

### 4.5. Optimal Transplant Site for Adipose Tissue-Derived Mesenchymal Stromal Cells in T1DM Cellular Therapy

The optimal transplantation site for ADMSCs needs to be addressed to avoid complications (e.g., pulmonary embolism) and provide the best response from the transplant. There is still no consensus as to the most suitable transplant site for cell therapy using MSCs for T1DM. The liver tended to be selected in past clinical trials [163,164] but the validity whether this organ was optimal for the delivery of ADMSCs has been raised. Intrahepatic transplantation can reduce immunological rejection [20], but has some potential disadvantages, such as severe inflammatory reaction against allogeneic islet grafts [165] and portal venous embolism of the MSCs themselves [14].

Yaochite et al. proposed an intrasplenic or intrapancreatic route for ADMSC transplantation [166]. They explained the advantages of the spleen as having fewer risks of severe complications, a potential for β-cell regeneration, and a reduced likelihood of graft rejection due to the promotion of immunotolerance [14,38,165]. In contrast, Bhang and colleagues achieved normoglycemia using a rodent model of subcutaneous hybrid islet and ADMSCs transplantation [109]. Subcutaneous tissue offers an ease of transplantation but also has the disadvantage of poor transplant efficacy due to the hypovascularity of the tissue. Transplantation of ADMSCs in subcutaneous tissue promotes angiogenesis and engraftment of the transplanted islets, which have intrinsically poor microvasculature. Further studies are required to identify an optimal transplantation site for ADMSCs.

## 5. Clinical Trials on Adipose Tissue-Derived Mesenchymal Stromal Cell Therapy for T1DM

In recent years, numerous clinical trials have tested the utility of ADMSCs in the treatment of various diseases (e.g., chronic ischemic cardiomyopathy, idiopathic pulmonary fibrosis, complex perianal fistula in Crohn’s disease, osteoarthritis) [167,168,169,170,171,172,173]. Comparatively, however, very few clinical trials have been conducted for the treatment of T1DM using ADMSCs. The clinical trials that have tested this treatment strategy showed positive outcomes (Table 3).

The first clinical trial was published in 2008 as a preliminary study [174]. The group performed transplantation of differentiated-IPCs derived from human ADMSCs using nicotinamide, activin A, exendin 4, pentagastrin, and HGF along with BM hematopoietic stem cells (HSCs) into five patients with T1DM. They found an increase in serum C-peptide levels, a decrease in the requirement for insulin, and no diabetic ketoacidosis or host immune response in these recipients [174]. In 2010, the group published the long-term outcomes of the treatment, after a mean follow-up of 7.3 months [163]. Similar to the results reported in the preliminary trial, the patients with T1DM (*n* = 11) showed increased serum C-peptide levels (0.02–0.1 ng/mL to 0.1–1.8 ng/mL), decreased insulin dependence (0.42–2.1 units/kg BW/day to 0.09–1.0 units/kg BW/day), and lower HbA1c levels (6.2–10.3% to 5.7–9.0%) [163].

In 2015, another two-armed prospective clinical trial was performed by the same group, expecting long-term engraftment, with the hope to be free from the need to use an immunosuppression reagent in the future [164]. In this study, approximately 2.7 × 10^4^ differentiatedIPCs per body weight (kg) from autologous or allogeneic ADMSCs with HSCs were transplanted into 20 patients with T1DM. Two years after the transplantation, the patients in the autologous group showed increased serum C-peptide levels (0.22 ± 0.21 ng/mL to 0.93 ± 0.24 ng/mL), decreased exogenous insulin requirements (63.90 ± 20.95 units/day to 39.66 ± 9.37 units/day) and lower HbA1c levels (10.99% ± 2.10% to 7.75% ± 1.05%). However, the allogeneic group also had increased serum C-peptide levels (0.028 ± 0.010 ng/mL to 0.460 ± 0.290 ng/mL), decreased exogenous insulin requirements (57.55 ± 21.82 units/day to 38.50 ± 13.34 units/day) and lower HbA1c levels (11.93% ± 1.90% to 8.01% ± 1.04%). The serum C-peptide levels were significantly better in the autologous group compared with the allogeneic group, and the authors claimed the superiority of autologous ADMSCs in terms of this long-term control of hyperglycemia [164].

Beside these clinical trials, the same group showed several reports using ADMSCs for T1DM treatment in patients with polyglandular autoimmune syndrome [175], terminal renal disease co-implanted with kidney [176], and chronic pancreatitis due to parathyroid adenoma [177]. Thus, the availability and safety of human ADMSCs for T1DM have been gradually clarified in the clinical setting. However, all of these clinical trials were undertaken by a single group and the transplant method was limited to the use of the hybrid transplantation of differentiated-IPCs with HSCs. Thus, these outcomes may contain bias.

Recently, novel clinical studies for the treatment of T1DM using human ADMSCs are ongoing by two other groups. Paspaliaris’s group in the Philippines has transplanted autologous activated-stromal vascular fractions to patients with T1DM without the use of immunosuppressants from 2007 to 2009 (NCT00703599; Phase I/II Study of Intravenous Administration of Activated Autologous Adipose-Derived Stromal Vascular Fraction in Patients with Type 1 Diabetes). An interim report showed that the treatment led to reduced insulin dependence, lower anti-hyperglycemic medication dosages, lower HbA1c levels, and increased serum C-peptide levels, without any evidence of organopathy [178]. This result indicates the clinical efficacy of ADMSCs rather than IPCs. In addition, a phase I clinical study in Jordan has been ongoing since 2017 (NCT02940418; The Use of Mesenchymal Stromal Cells (MSC) in Type 1 Diabetes Mellitus in Adult Humans: Phase I Clinical Trial). In this two-armed study, allogeneic ADMSCs and autologous HSCs are intravenously injected into 20 candidates in two discrete cellular dosages. No immunosuppressants have been used. The outcomes have yet to be published but may indicate the proper numbers of ADMSCs required for treatment success.

## 6. Future Perspectives

At the moment, islet transplantation is the only reliable cellular replacement therapy that can reverse T1DM [12]. However, its utility in the clinic is plagued by poor transplant efficacy and a shortage of suitable donors [12,14,165]. In this regard, ADMSCs are representative candidates for biological materials which are being tested as a way to overcome these disadvantages. Over the years, several types of hybrid transplantation systems combining two different cellular lineages (e.g., islets and MSCs, MSCs and HSCs) have been used for T1DM treatment in humans and in animals, with most studies showing the superiority of the combination compared with the use of MSCs alone [95,98,105,106,107,109,179]. Thus, hybrid islet transplantation combined with ADMSCs is expected as a future human therapy for T1DM, a practical cellular replacement therapy offering the potential to normalize hyperglycemia, as noted by the reported successful outcomes in many animal experiments [105,109,112].

Due to their abundancy, authors can afford to use ADMSCs in large quantities and any number of times. With this in mind, we conceived a multi-cellular combinational transplantation strategy for treating T1DM using ADMSCs pre-treated to have a specific function: (1) insulin-producing ADMSCs; (2) immunomodulatory ADMSCs; and (3) regenerative ADMSCs. For this to work, first, an insulin-producing cell needs to be created from ADMSCs in vitro. ADMSCs have the potential to differentiate into IPCs, with proliferation and the insulin-producing capacity promoted via culturing [78,180] or genomic editing, such as overexpressing PDX1 [77,80]. Next, immunomodulation-specific ADMSCs should be established, which will protect resident β-cells and the newly transplanted differentiated-IPCs from autoimmunity or graft rejection. ADMSCs overexpressing *TGF-β* and *THRB1* may reinforce the cellular immunosuppressive potential [181]. Moreover, the use of the alginate and hyaluronic acid hydrogel scaffolds during transplantation may help to maintain the immunosuppressive capacity [96]. Finally, it is necessary to prepare revascularization-specific regenerative ADMSCs, which can restore a suitable niche for differentiated-IPCs. Given that ADMSCs secrete angiogenic molecules such as VEGF, IGF, HGF [61,62,182], gene-engineering to overexpress these growth factors in the transplanted ADMSCs may be an ideal strategy. Guadalupe and colleagues reported that enhanced neovascularization was achieved in myocardial infarction models by transplanting porcine ADMSCs overexpressing *Igf* and *Hgf* [183]. Furthermore, the homing potential of ADMSCs is also reinforced via bioengineering to overexpress the C-X-C chemokine receptor type 4 [184]. To perform the treatment using ADMSCs effectively, we consider the two methodological strategies. Regarding the timing of transplantation, the strategy that regenerative ADMSCs supplement prior to the differentiated-IPCs could be effective because it requires a span for angiogenesis [116]. Regarding the procedure of transplantation, the liver or subcutaneous tissues may be reasonable transplant sites due to their accumulating capacity and easy procedures like percutaneous infusion under the local anesthesia. The patients can acquire gratifying therapeutic outcomes due to the additional transplantation of ADMSCs using this procedure in case that the therapeutic effect was unsatisfied by one transplantation.

To the best of our knowledge, multi-cellular transplantation combined with different function-enhanced ADMSCs has not been reported. However, we consider that these combination therapies are theoretically feasible and may offer a promising T1DM treatment strategy in the future.

## 7. Conclusions

This review explored the potential of ADMSCs for the treatment of T1DM. While there remains a paucity of literature, including any evidence of the long-term outcomes and suitability of transplant procedures, it is clear that ADMSCs are advantageous in their abundance, easiness of acquirement, and prominent multipotentiality. ADMSCs offer an alternative to β-cells and can aid in the reversal of T1DM.

## Figures and Tables

**Figure 1 jcm-08-00249-f001:**
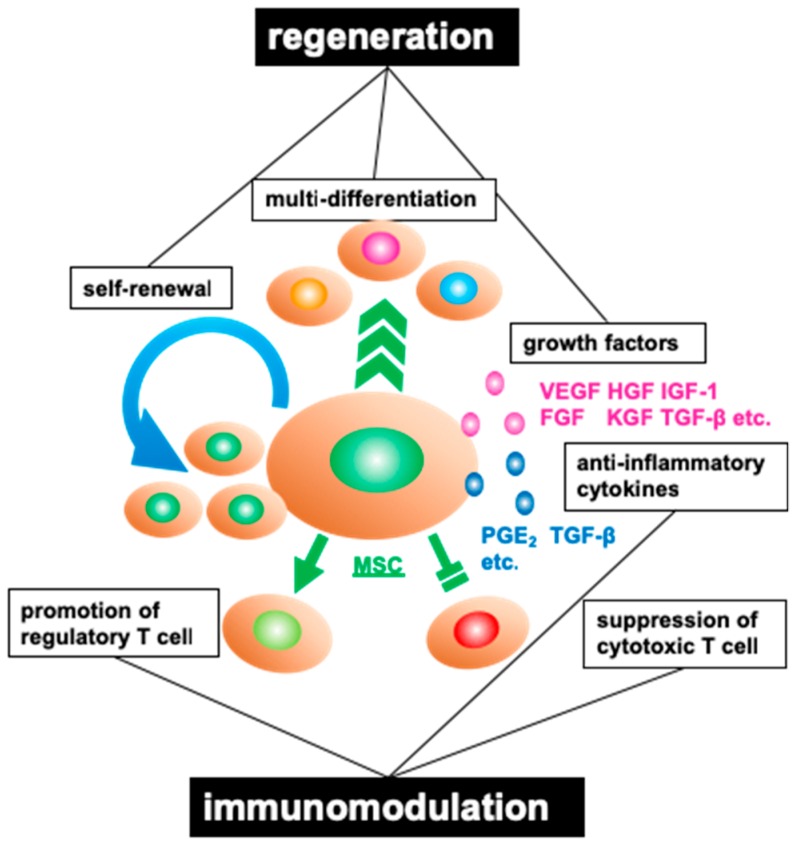
Major properties of mesenchymal stromal cells (MSCs). MSCs are capable of self-renewal, regeneration multi-differentiation, and are endowed with immunomodulatory potential. These cells express various growth factors (e.g., VEGF, vascular endothelial growth factor; HGF, hepatocyte growth factor; IGF-1, insulin-like growth factor-1; FGF, fibroblast growth factor; KGF, keratinocyte growth factor; TGF-β, transforming growth factor-β) and anti-inflammatory cytokines (e.g., PGE2, prostaglandin E2). MSCs also suppress the activities of cytotoxic T cells and promote the production of regulatory T cells.

**Figure 2 jcm-08-00249-f002:**
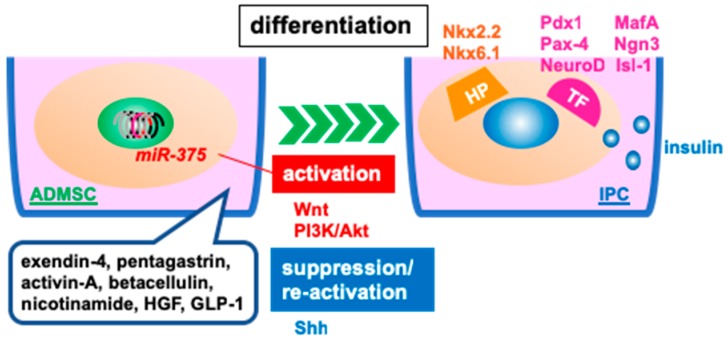
Differentiated insulin-producing cells (IPCs) for transplantation. Adipose tissue-derived mesenchymal stromal cells (ADMSCs) are differentiated into IPCs in the culture medium supplemented with exendin-4, pentagastrin, activin-A, betacellulin, nicotinamide, hepatocyte growth factor (HGF), and glucagon-like peptide-1 (GLP-1). Wnt and the phosphoinositide 3-kinase (PI3K) signaling pathway play a role in the activation of differentiation, whereas the sonic hedgehog (Shh) signaling pathway works in suppression and re-activation. MicroRNA-375 (miR-375) also promotes differentiation. The differentiated-IPCs express some homeobox proteins (HPs), including Nkx2.2 and Nkx6.1; and transcription factors (TFs), including Pdx-1, MafA, Pax-4, Ngn3, NeuroD and Isl-1.

**Figure 3 jcm-08-00249-f003:**
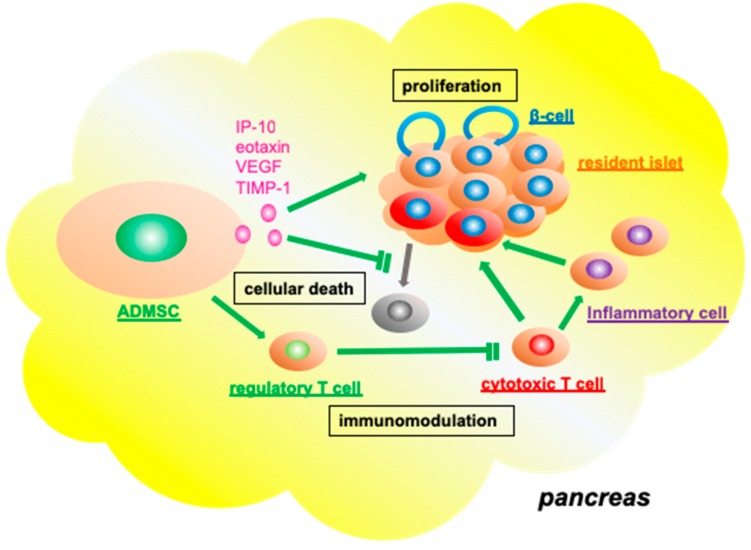
Functional role of adipose tissue-derived mesenchymal stromal cells in the resident pancreatic islets. Transplanted adipose tissue-derived mesenchymal stromal cells (ADMSCs) release functional molecules, including interferon gamma-induced protein-10 (IP-10), eotaxin, vascular endothelial growth factor (VEGF), and tissue inhibitor of metalloproteinase-1 (TIMP-1) to promote the viability and proliferative capacity of endogenous β-cells. ADMSCs also upregulate regulatory T cells and downregulate cytotoxic T cells, which inhibit inflammatory cell infiltration (immunomodulation).

**Figure 4 jcm-08-00249-f004:**
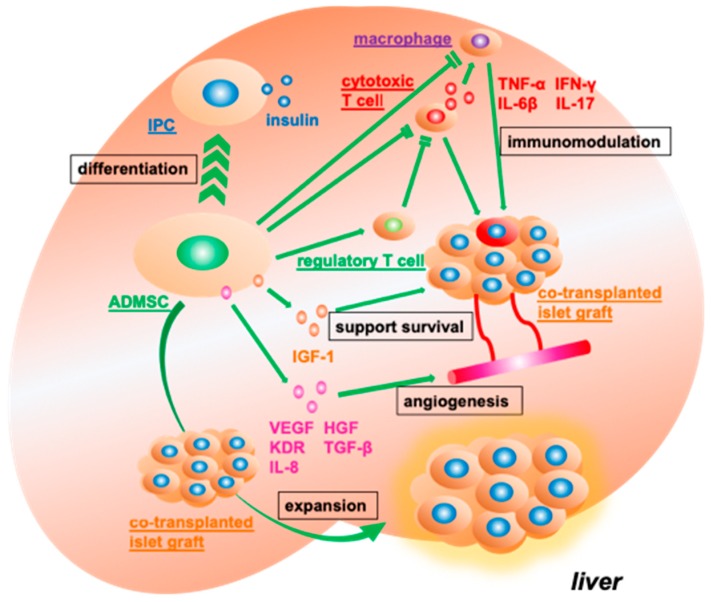
Supporting the function and engraftment of co-transplanted islet grafts. When adipose tissue-derived mesenchymal stromal cells (ADMSCs) are transplanted with islets, the cells can contribute to improving type 1 diabetes mellitus (T1DM) via several mechanisms. ADMSCs release functional molecules, including vascular endothelial growth factor (VEGF), hepatocyte growth factor (HGF), kinase insert domain receptor (KDR), transforming growth factor-β (TGF-β), interleukin-8 (IL-8), and insulin-like growth factor-1 (IGF-1) to promote angiogenesis and support the survival of co-transplanted islet grafts. ADMSCs upregulate regulatory T cells and downregulate cytotoxic T cells and macrophages. Inflammatory cytokines, including tissue necrosis factor-α (TNF-α), interferon gamma (IFN-γ), interleukin-6β (IL-6β), and IL-17, are suppressed in the presence of ADMSCs. In addition, ADMSCs can differentiate into insulin-producing cells (IPCs) and expand co-transplanted islet grafts.

**Figure 5 jcm-08-00249-f005:**
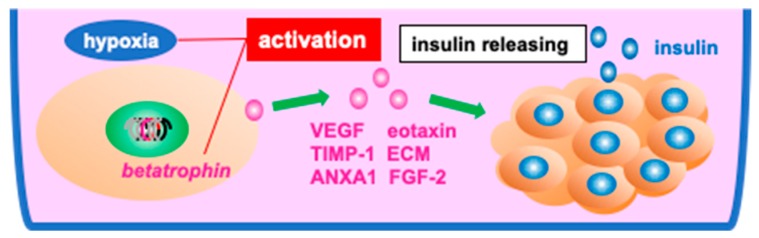
Supporting the function of cultured islet grafts for transplantation. Preculturing islet grafts with adipose tissue-derived mesenchymal stromal cells (ADMSCs) before transplantation enhances the insulin-releasing potential of the islet graft. ADMSCs support the islet in culture via the expression of and cross talk from vascular endothelial growth factor (VEGF), eotaxin, tissue inhibitor of metalloproteinase-1 (TIMP-1), extracellular matrix (ECM) components, annexin A1 (ANXA1), and fibroblast growth factor-2 (FGF-2). In addition, overexpression of betatrophin gene together with hypoxic culture conditions of islets graft with ADMSCs may enhance graft transplantation efficiency.

**Table 1 jcm-08-00249-t001:** Representative studies pertaining to insulin-producing cells (IPCs) differentiated from adipose tissue-derived mesenchymal stromal cells (ADMSCs).

Author	Year	Ref.	Donor of ADMSCs Species	Source	Procedure of Differentiation or Transplantation	Outcomes
Timper	(2007)	[74]	Human	Uncertain	ADMSCs were cultured in serum-free medium with exendin-4, pentagastrin, activin-A, betacellulin, nicotinamide, and HGF. (No transplantation)	Expressed INSULIN, GLUCAGON and SOMATOSTATIN. Potential of insulin secretion was not shown.
Okura	(2009)	[75]	Human	Omentum	ADMSCs were cultured following a five-step method for the differentiation of ESCs into IPCs. (No transplantation)	Detected insulin and C-peptide.
Kang	(2009)	[76]	Human	Eyelid	ADMSCs were cultured in medium containing serum, nicotinamide, activin and/or GLP-1, then differentiated into IPCs. 1.5 × 10^6^ cells were transplanted beneath the kidney capsules of STZ treated-immunodeficient mice. (Xenotransplantation)	Secreted insulin and C-peptide under glucose stimulation. 50% of transplanted mice achieved normoglycemia.
Kajiyama	(2010)	[77]	Mice	Inguinal fat	5.0 × 10^5^ ADMSCs transferred *pdx-1* were infused into the tail vain of STZ treated-mice. (Syngeneic transplantation)	Potential of insulin secretion was not shown. Decreased blood glucose levels and increased survival.
Chandra	(2011)	[78]	Human	Abdomen	ADMSCs were cultured in the medium with serum, insulin, transferrin, selenium, activin A, sodium butyrate, FGF, GLP-1, nicotinamide and non-essential amino acids, then differentiated into IPCs. The 1000–1200 cells packed in immuno-isolatory capsules were infused into the peritoneal cavities of STZ treated-mice. (Xenotransplantation)	Produced human C-peptide under glucose stimulation. Reduced blood glucose levels. No achievement of normoglycemia.
Kim	(2012)	[79]	Human	Uncertain	Compared growth potential of ADMSCs, BM-MSCs, umbilical cord-derived and periosteum-derived MSCs into IPCs in vitro. (No transplantation)	Only periosteum derived-MSC showed a response in glucose concentration.
Lee	(2013)	[80]	Human	Abdomen	2.0 × 10^6^ ADMSCs expressing PDX-1 were transplanted into the kidney capsule of STZ treated-immunodeficient mice. (Xenotransplantation)	Exhibited insulin secretion in response to glucose. Reduced blood glucose levels. No achievement of normoglycemia.
Nam	(2014)	[81]	Human	Eyelid	ADMSCs were differentiated into IPCs using a commercial medium. 1.5 × 10^6^ cells were transplanted into the kidney capsules of low STZ and insulin treated-immunodeficient mice. (Xenotransplantation)	Secreted insulin and C-peptide under glucose stimulation. Reduced blood glucose levels. No achievement of normoglycemia.
Sun	(2017)	[82]	Human	Uncertain	1.0 × 10^6^ ADMSCs overexpressing BETATROPHIN were infused into the tail vein of STZ treated-mice. (Xenotransplantation)	Promoted proliferation and insulin release in co-culture islets. Decreased blood glucose levels significantly better than in the control group.
Amer	(2018)	[83]	Rat	Abdomen	ADMSCs were cultured in the medium with serum, activin A, exendin 4, pentagastrin, HGF, and nicotinamide, then differentiated into IPCs. 1.5 × 10^6^ cells were infused into the splenic artery of STZ-treated rats. (Syngeneic transplantation)	Expressed β-cell markers and secreted insulin. Showed apparent regeneration, diffuse proliferation of resident islets and increased serum insulin levels. Achieved normoglycemia.

Abbreviations: ADMSCs, adipose tissue-derived MSCs; ESCs, embryonic stem cells; FGF, fibroblast growth factor; GLP-1, glucagon-like peptide-1; HGF, hepatocyte growth factor; MSCs, mesenchymal stromal cells; STZ, streptozotocin.

**Table 2 jcm-08-00249-t002:** Representative studies using adipose tissue-derived mesenchymal stromal cells (ADMSCs) in hybrid islet transplantation for diabetes.

Authors	Year	Ref.	Donor of ADMSCs Species	Source	Number	Donor of Islets Species	Number	Procedure of Transplantation	Outcomes
Ohmura	(2010)	[105]	Mice	Inguinal fat	2 × 10^5^ cells	Mice	200 islets	Renal subcapsular transplantation into STZ-treated mice. (Syngeneic transplantation)	Reversed diabetes status and prolonged islet graft survival. Suppressed CD4+/CD8+ T cells.
Cavallari	(2012)	[108]	Human	Subcutaneous fat	2.5 × 10^4^ cells	Rat	500 islets	Intrahepatic transplantation into STZ-treated rats. (Xenotransplantation)	Achieved better glycemic control as compared with islet transplantation alone.
Karaoz	(2013)	[95]	Rat	Peritoneal fat	1.0 × 10^6^ cells	Rat	500 islets	Compared transplant efficacy among islet alone or islet with ADMSCs or islet with BM-MSCs into the kidney capsule of STZ-treated rats. (Syngeneic transplantation)	Co-transplanted with ADMSCs rats showed the greatest efficacy.
Bhang	(2013)	[109]	Human	Uncertain	8 × 10^5^ cells	Rat	800 islets	Transplanted with FGF-2 into the dorsal subcutaneous area of STZ-treated mice. (Xenotransplantation)	Achieved normoglycemia. The therapeutic effect was enhanced by addition of FGF2.
Mohammadi	(2017)	[107]	Mice (C57BL/6)	Abdomen	2 × 10^5^ cells	Mice (BALB/c)	200 islets	Transplanted with hydrogel into the intraperitoneal spaces of STZ-treated mice. (Allogeneic transplantation)	Decreased pro-inflammatory cytokines and increased Treg. Increased transcript levels in *Iod, Inos* and *Pdx1* in the presence of ADMSCs.
Song	(2017)	[113]	Human	Abdomen	1 × 10^4^ cells	Mice	125–150 islets	Renal subcapsular transplantation into STZ-treated mice. After chronic pancreatitis surgery. (Xenotransplantation)	Improved islet survival and function. Showed IGF-1 secretion, suppression of inflammation, and promotion of angiogenesis.
Navaei	(2018)	[98]	Human	Epididymal	6 × 10^6^ cells	Rat	1000 IEQs	Intra-omental transplantation to STZ treated-mice. (Xenotransplantation)	Significantly promoted survival, engraftment and insulin production.
Tanaka	(2018)	[106]	Mice	Inguinal fat	1.0 × 10^5^, 5.0 × 10^5^ or 1.0 × 10^6^ cells	Mice	50 islets	Renal subcapsular transplantation to STZ treated-mice. (Syngeneic transplantation)	Expanded islet graft resulted in ameliorating hyperglycemia.

Abbreviations: ADMSCs, adipose tissue-derived MSCs; BM-MSCs, bone marrow-derived MSCs; FGF, fibroblast growth factor; IGF-1, insulin-like growth factor-1; MSCs, mesenchymal stromal cells; STZ, streptozotocin; Treg, T regulatory cells.

**Table 3 jcm-08-00249-t003:** Diabetes clinical trials using adipose tissue-derived stromal cells (ADMSCs).

Authors/(Year)		Number of Patients	Age	year	Disease Duration/(year)	Number of ADMSCs	Pre/Post Infusion	C-peptide/(ng/mL)	HbA1c/(%)	Insulin Requirement/(Units/day)	Follow-Up/(Months)
Vanikar [163]/(2010)		11	21.1	(13–43)	8.2/(1–24)	3.0 × 10^6^	Pre	0.10/(0.02–0.30)	8.47/(6.22–10.30)	1.14/kg BW/(0.42–2.10)	7.3/(2.2–12.0)
							Post	0.37/(0.1–1.8)	7.39/(5.72–8.98)	0.63/kg BW/(0.09–1.00)	
Thakkar [164]/(2015)	Auto-	10	20.20 ± 6.90		8.1 ± 3.4	2.7 ± 0.8 × 10^2^ (/μL) × 103.1 ± 28.3 (mL)	Pre	0.220 ± 0.210	10.99 ± 2.10	63.90 ± 20.95	33.10 ± 18.38
							Post (2y)	0.930 ± 0.240	7.75 ± 1.05	39.66 ± 9.37	
	Allo-	10	19.70 ± 9.96		9.9 ± 7.1	2.1 ± 0.7 × 10^2^ (/μL) × 95.3 ± 14.2 (mL)	Pre	0.028 ± 0.010	11.93 ± 1.90	57.55 ± 21.82	54.24 ± 15.75
							Post (2y)	0.460 ± 0.290	8.01 ± 1.04	38.50 ± 13.34	

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
