# Peer review of "Regenerative and Transplantation Medicine: Cellular Therapy Using Adipose Tissue-Derived Mesenchymal Stromal Cells for Type 1 Diabetes Mellitus"

_jcm, 2019, doi:10.3390/jcm8020249_

Reviewer 1 Report

Overall the manuscript provides an interesting Review of the possible cell sources for T1D cell therapy and in particular mesenchymal stem cells derived from adipose tissue. The subject is interesting for the scientific community as well as for T1D patients. However, the English language used in the manuscript is uneven and due to the lack of a clear and succinct expression in the use of the English language the reader will find it difficult to appreciate the full content of the Manuscript. An extensive amelioration of the English language, grammar and syntaxes, will increase the quality of the Manuscript.

Some parts, especially at the end of the chapters, require a better summary of the observations and data reported by the authors. This will ameliorate substantially the manuscript.

Although not extensive, below are few suggestions and comments to the authors for ameliorating the manuscript:

1. In the Introduction section it should be specified that T1D is only a small percentage (5%) of the DM patients.

2. Lines 54-58: It should be noted that pancreas transplantation represents an interest, especially for type 1 diabetes patients with end-stage renal disease and hypoglycaemic unawareness. This therapy represents post-operational risk for complications such as thrombosis, pancreatitis, infection bleeding and rejection (doi:10.2337/diacare.26.12.3288).

3. Line 62: Instead of “We consider …” maybe it should be reformulated as: “We review herein the possibility of an alternative therapy that is stem cell therapy”.

4. Line 71: Properties of Mesenchymal stem cells instead of "Background…

5. Lines of paragraph 2.2: It is needed to add clarity in the description. For example on line 129, which important DC molecules are inhibited?

6. Lines 140-142: How ADMSC transplantation might contribute to T1D? Through which mechanism? Conclusions have to be made by proposing briefly at least some relevant hypothesis.

7. Line 216: 3.3 Subtitle suggestion:  Instead of Supporting the function and proliferation of resident pancreatic islets one proposition for a better comprehension could be:”Functional role of ADMSCs in the resident pancreatic islets” or “ADMSCs contribute to beta cell differentiation and prevent beta cell death”. Similar suggestion applies for the heading of legend of Fig. 3

8. Line 300: Subitle suggestion: 3.5 In vitro co-culture of syngeneic islets grafts with ADMSCs ameliorates transplantation efficiency".

9. Line 317: Suggestion: “Overexpression of betatrophin gene similarly improves… instead of “Genomically overexpression of ….” 

Similarly on line 326 a suggestive correction could be “In addition, overexpression of betatrophin gene together with hypoxic culture conditions of islets grafts with ADMSCs may enhance graft transplantation efficiency.

10. Line 334: Suggestion for Heading 4. “Towards a suitable adipose tissue-derived mesenchymal stem cell transplantation”.

11. Line 335: Suggestion:  “Sources of adipose tissue-derived mesenchymal stem cells

12. Line 350: Suggestion for Heading 4.2: “Methods for the preparation of adipose tissue-derived mesenchymal stem cell”.

13. Line 395: Suggestion: Taken together the above described observations indicate that subcutaneous….

14. Lines 404-406: Language has to be corrected. The sentence does not make good sense

15. Similarly lines 440-441

16. Names of genes and proteins have to be in conformity to the accepted nomenclature rules. For example TGF-b for the human gene and TGF-b for the protein (use italic for a symbol when the gene is meant and plain (roman) for when the protein is meant). Human genes are in capital letters etc.

17. Lanes 536-538: Conclusions have to be reformulated by briefly adding a summary of the advantages and pitfalls of the ADMSCs for T1D therapy and by taking in consideration the English language.

Finally a minor concern is the accuracy of the subject as described in the title. One proposition is to reformulate the title as follows: 

“Regenerative and Transplantation Medicine: Cellular Therapy Using Adipose Tissue-Derived Mesenchymal Stem Cells for Type 1 Diabetes Mellitus”

Author Response

Thank you for your kind peer reviews for our review article. As shown below, we revised our manuscript following your suggestions. These valuable suggestions improve the quality of our this work. 

Point 1:In the Introduction section it should be specified that T1D is only a small percentage (5%) of the DM patients.

Response 1: Thank you for suggestion. The information about the population of T1D is inserted in the revised Introduction (Page 1, Line 39-40).

Point 2:Lines 54-58: It should be noted that pancreas transplantation represents an interest, especially for type 1 diabetes patients with end-stage renal disease and hypoglycaemic unawareness. This therapy represents post-operational risk for complications such as thrombosis, pancreatitis, infection bleeding and rejection (doi:10.2337/diacare.26.12.3288).

Response 2:We include the benefits and risks of pancreas transplantation in Introduction following your suggestion.The new reference is also inserted. (Page 2, Line 54-57)

Point 3:Line 62: Instead of “We consider …” maybe it should be reformulated as: “We review herein the possibility of an alternative therapy that is stem cell therapy”.

Response 3:We reformulated them as you mentioned.(Page 2, Line 65-66)

Point 4:Line 71: Properties of Mesenchymal stem cells instead of "Background…

Response 4: We reformulated them following your recommendation.(Page 2, Line 74)

Point 5:Lines of paragraph 2.2: It is needed to add clarity in the description. For example on line 129, which important DC molecules are inhibited?

Response 5: Expression of CD80, CD83 and CD86 are downregulated on DCs in the presence of ADMSC. We include the information in revised version. (Page 4, Line 131-132)

Point 6:Lines 140-142: How ADMSC transplantation might contribute to T1D? Through which mechanism? Conclusions have to be made by proposing briefly at least some relevant hypothesis.

Response 6: Thank you for suggestion. We added the expected mechanisms following: Their superior immunomodulatory and anti-inflammatory functions may contribute to prevention of graft rejection, and the better trophic effect support engraftment of transplanted islets. (Page 4, Line 145-147)

Point 7:Line 216: 3.3 Subtitle suggestion:  Instead of Supporting the function and proliferation of resident pancreatic islets one proposition for a better comprehension could be:”Functional role of ADMSCs in the resident pancreatic islets” or “ADMSCs contribute to beta cell differentiation and prevent beta cell death”. Similar suggestion applies for the heading of legend of Fig. 3

Response 7:Thank you for suggestion, we changed to recommended version. The legend of Fig. 3 is also changed. (Page 8, Line 221; Page 8, Line 233)

Point 8:Line 300: Subtitle suggestion: 3.5 In vitro co-culture of syngeneic islets grafts with ADMSCs ameliorates transplantation efficiency".

Response 8:We changed the subtitle as you suggested. (Page 12, Line 305-306)

Point 9:Line 317: Suggestion: “Overexpression of betatrophin gene similarly improves… instead of “Genomically overexpression of ….” 

Similarly on line 326 a suggestive correction could be “In addition, overexpression of betatrophin gene together with hypoxic culture conditions of islets grafts with ADMSCs may enhance graft transplantation efficiency.

Response 9:We changed the sentences as you suggested. (Page 12, Line 323-324; Page 13, Line 332-334)

Point 10:Line 334: Suggestion for Heading 4. “Towards a suitable adipose tissue-derived mesenchymal stem cell transplantation”.

Response 10:We changed the Heading 4 as you suggested. (Page 13, Line 341)

Point 11:Line 335: Suggestion:  “Sources of adipose tissue-derived mesenchymal stem cells”

Response 11:Changed. (Page 13, Line 342) 

Point 12:Line 350: Suggestion for Heading 4.2: “Methods for the preparation of adipose tissue-derived mesenchymal stem cell”.

Response 12:Thank you, we changed. (Page 13, Line 357)

Point 13:Line 395: Suggestion: Taken together the above described observations indicate that subcutaneous….

Response 13:We changed the sentence as you suggested. (Page 14, Line 402)

Point 14:Lines 404-406: Language has to be corrected. The sentence does not make good sense

Response 14:We revised the sentence as following: At present, cancer therapeutic strategiesdepend on the genesis and stage [150]. Therefore, we should identify the origin and prepare for the indeterminate occurrence of stromalcell-derived neoplasms. (Page 14, Line 412-414)

Point 15:Similarly lines 440-441

Response 15:We revised the sentence, “Further studies are required to identify an optimal transplantation site for ADMSCs.” (Page 15, Line 448)

Point 16:Names of genes and proteins have to be in conformity to the acceptednomenclature rules. For example TGF-bfor the human gene and TGF-b for the protein (use italic for a symbol when the gene is meant and plain (roman) for when the protein is meant). Human genes are in capital letters etc.

Response 16: We checked all of the words of genes and proteins in this review, and corrected them in appropriate form. (Page 5, Table 1; Page 6, Table 1; Page 7, Line 196; Page 10, Table 2; Page 17, Line 520; Page 18, Line 523; Page 18, Line 530-531)

Point 17:Lanes 536-538: Conclusions have to be reformulated by briefly adding a summary of the advantages and pitfalls of the ADMSCs for T1D therapy and by taking in consideration the English language.

Response 17:We reformulated Conclusion section by adding the summary. (Page 18, Line 546-548)

Point 18:Finally a minor concern is the accuracy of the subject as described in the title. One proposition is to reformulate the title as follows: 

“Regenerative and Transplantation Medicine: Cellular Therapy Using Adipose Tissue-Derived Mesenchymal Stem Cells for Type 1 Diabetes Mellitus”

Response 18: We reformulated the main title as you suggested. (Page 1, Title)

Appendix:This manuscript was received English proofreading bya native English speaker (Ms. Rebecca Jackson, PhD, from Edanz Group) before the first submission. Your precise advices enable the manuscript to be brushed up. Thank you.

Reviewer 2 Report

The review paper written by Takahashi et al., describes the potential therapeutic application of Adipose tissue derived Mesenchymal Stem Cells in diabetes type 1.

Overall comment:

Reviewer feels that paper is well written and would be a significant support for other researchers who are interested in MSCs and their therapeutic application.

However the reviewer is also not in align with some statements in the manuscript (please see in specific comment). 

Specific Comment: 

1) Authors does not really address what is the "special" about MSCs derived from Adipose tissue and other sources and in which aspects AdMSCs are superior to BMMSCs. Moreover, Authors did not mention the benefit of taking MSC from fat tissue e.g., the potential differences in allogeneic vs autologous. Simply, to obtain adipose tissue from patient requires much less manipulations compared to Bone Marrow. 

2) The word "stem" is confusing since true stemness of the MSCs is contradictory. It is better to use Mesenchymal Stromal Cells. 

3) in line 84-85 authors describe the advantage of MSC over iPSCs and ESCs it terms of ethical concerns. The reviewer is not sure to what ethical concerns authors applying to in terms of iPSCs since these cells are originally derived from human skin fibroblasts of adult person. 

Author Response

Thank you for your kind peer review. Following your valuable suggestions, we revised our manuscript.

Point 1: Authors does not really address what is the "special" about MSCs derived from Adipose tissue and other sources and in which aspects AdMSCs are superior to BMMSCs. Moreover, Authors did not mention the benefit of taking MSC from fat tissue e.g., the potential differences in allogeneic vs autologous. Simply, to obtain adipose tissue from patient requires much less manipulations compared to Bone Marrow. 

Response 1: This review focuses on ADMSCs in particular, therefore we did not describe the speciality of MSC itself in detail. As you know, this manuscript contains a huge of information, and additional long descriptions may impair reader’s understanding. Therefore, we mentioned the necessity minimum about MSCs characteristics in Section 2.1. (Page 2, Line 74-105)

On the other hand, we actually described the superiority of ADMSCs to BM-MSCs in Section 2.2, for example, proliferative capacity, immunomodulation, trophic effects and etc. (Page3, Line 121-122; Page 4, Line 126-133; Page 4, Line 137-142).The in vivotransplantation study directly compared ADMSCs with BM-MSCs for DM treatment was limited, but Karaoz et al. revealed the superiority of ADMSCs co-transplantation with islet grafts (Page 9, Line 285-287; Page 10, Table 2). 

As you mentioned, one of the benefits using ADMSCs is the easy isolation procedure (and abundance). The isolation efficacy is approximately 500-times the yield of stromal cells attained from the bone marrow. We described these terms in Section 4.2. (Page 13, Line 361-367). However, unfortunately, we could not find references which describe the differences in allogeneic vs autologous ADMSCs.

Point 2: The word "stem" is confusing since true stemness of the MSCs is contradictory. It is better to use Mesenchymal Stromal Cells. 

Response 2: Thank you for your suggestion. Almost all the “stem” words were changed to “stromal”. (Page 1, Line 4; Page 1, Line 26; Page 1, Line 31; Page 2, Line 67; Page 2, Line 73; Page 2, Line 74; Page 2, Line 85; Page 3, Line 90; Page 3, Line 108; Page 3, Line 112; Page 3, Line 114; Page 4, Line 148; Page 4, Line 150; Page 4, Line 160; Page 5, Line 170; Page 6, Line 171; Page 7, Line 181; Page 8, Line 221; Page 8, Line 233; Page 8, Line 234; Page 10, Line 290; Page 11, Line 291; Page 12, Line 295; Page 12, Line 305; Page 13, Line 329; Page 13, Line 341; Page 13, Line 342; Page 13, Line 357; Page 13, Line 358; Page 14, Line 368; Page 14, Line 407; Page 14, Line 414; Page 15, Line 432; Page 15, Line 450; Page 16, Line 469)

Point 3: in line 84-85 authors describe the advantage of MSC over iPSCs and ESCs it terms of ethical concerns. The reviewer is not sure to what ethical concerns authors applying to in terms of iPSCs since these cells are originally derived from human skin fibroblasts of adult person. 

Response 3: Exactly.The ethical problem associated with iPSCs is similar to ADMSCs. We deleted the statement about iPSCs. (Page 2, Line 87-88)

Round  2

Reviewer 1 Report

I thank the authors for their reply and for considering the proposed suggestions. Their Manuscript provides an interesting review of the current state of cell therapy for Type 1 diabetes by using adipose tissue-derived mesenchymal stromal cells.

Just one minor comment concerning the nomenclature of genes and proteins. The previous suggestion, on point 16 of the review, concerns the abbreviations of genes and proteins. While the authors have appropriately corrected the nomenclature in the Manuscript, on line 843 thrombospondin-1 should not appear in italic letters unless the abbreviation is used for the gene: THRB1 but not for the protein THRB1 (non-italic).

Response: Have revised accordingly.